# Advances in Sol-Gel-Based Superhydrophobic Coatings for Wood: A Review

**DOI:** 10.3390/ijms24119675

**Published:** 2023-06-02

**Authors:** Yudong Wang, Shangjie Ge-Zhang, Pingxuan Mu, Xueqing Wang, Shaoyi Li, Lingling Qiao, Hongbo Mu

**Affiliations:** College of Science, Northeast Forestry University, Harbin 150040, China

**Keywords:** wood, superhydrophobicity, sol-gel method, micro-nanomaterials

## Abstract

As the focus of architecture, furniture, and other fields, wood has attracted extensive attention for its many advantages, such as environmental friendliness and excellent mechanical properties. Inspired by the wetting model of natural lotus leaves, researchers prepared superhydrophobic coatings with strong mechanical properties and good durability on the modified wood surface. The prepared superhydrophobic coating has achieved functions such as oil-water separation and self-cleaning. At present, some methods such as the sol-gel method, the etching method, graft copolymerization, and the layer-by-layer self-assembly method can be used to prepare superhydrophobic surfaces, which are widely used in biology, the textile industry, national defense, the military industry, and many other fields. However, most methods for preparing superhydrophobic coatings on wood surfaces are limited by reaction conditions and process control, with low coating preparation efficiency and insufficiently fine nanostructures. The sol-gel process is suitable for large-scale industrial production due to its simple preparation method, easy process control, and low cost. In this paper, the research progress on wood superhydrophobic coatings is summarized. Taking the sol-gel method with silicide as an example, the preparation methods of superhydrophobic coatings on wood surfaces under different acid-base catalysis processes are discussed in detail. The latest progress in the preparation of superhydrophobic coatings by the sol-gel method at home and abroad is reviewed, and the future development of superhydrophobic surfaces is prospected.

## 1. Introduction

Population growth and the rapid progress of science and technology have caused a serious shortage of energy on Earth [1,2]. As a green and sustainable clean material in the environment, wood has been favored by people [3,4], and it is widely used in building materials, decorative materials [5,6,7], etc. In addition, wood has the advantages of being light [8,9,10], having large reserves [11,12,13], having excellent mechanical properties [14,15,16], and other advantages. The anisotropic nature of the vertically aligned channels in its internal microstructure [17,18] gives the wood a certain thermal insulation potential [19,20], which makes it have the potential for multifunction [21]. Such as high-density flame retardant wood [22,23], wood solar steam generator [24,25], transparent wood [26], etc. However, wood is a hygroscopic material [27,28]. It is subject to wetting [29,30] and dry shrinkage [31,32], resulting in deformation [33,34,35], corrosion [36,37], and a shortened service life under natural conditions. Therefore, the modification of wood to give it good hydrophobic properties has become the focus of current research.

Most of the modifications used to create superhydrophobic surfaces for wood come from thinking about nature. Such as lotus leaves floating on water [38,39,40], butterfly wings [41,42,43], and rose petals [44,45,46], are all super-hydrophobic surface structures that organisms adapt to nature. In the lotus leaf effect, the static contact angle between the lotus leaf surface and water is more than 150°, while the water sliding angle is only 1.9°. The water droplets attached to the surface of the lotus leaf can easily slide down and take away the pollutant particles on the surface, resulting in a self-cleaning effect [47,48]. Superhydrophobic surfaces in a static self-wetting state in nature provide an important reference for the construction of biomimetic wood superhydrophobic coatings [49,50,51].

The method of superhydrophobic modification on wood surfaces is based on the physical structure of the selected materials, mainly aiming at the construction of surface roughness [52,53] and the modification of low surface energy substances [54]. By observing the superhydrophobic surfaces of living organisms in nature, imitating the relevant characteristics of their adaptive functions and responses to the environment, and establishing micro-nanostructures with appropriate roughness on wood surfaces, artificial preparation of superhydrophobic coatings can be achieved [55,56,57]. Among them, the functional coatings include selective water-repellent superhydrophobic coatings for antivirus and antibacterial [58,59], superhydrophobic coatings for wood surfaces with self-cleaning and anti-icing effects [60,61], superhydrophobic coatings for oil-water separation and efficient separation of oil spills at sea in industrial production [62], and superhydrophobic coatings for solar panels with anti-reflection and self-cleaning functions [63,64]. The introduction of carbon nanotubes into the superhydrophobic coating can achieve good thermal management ability. Zhu et al. added PDMS-modified carbon nanotubes to wood aerogels to prepare oil-water separation wood aerogels based on the photothermal effect [65]. Wang et al. obtained a wood sponge material with low energy consumption and high oil absorption capacity by constructing Ti_3_C_2_T_x_ MXene nanosheets coated on the surface of a natural wood sponge and decorating it with polydimethylsiloxane, which has a broad development prospect in oil spill remediation [66]. In addition to the development of oil-water separation, the modification process of superhydrophobic coatings can be combined with self-healing materials. Team Li synthesized a superhydrophobic coating made of flower-like ZnO particles coated with modified poly-dopamine (PDA), which has excellent photothermal conversion performance. The prepared composite coating can achieve rapid healing ability lasting for 30 s in response to NIR stimulation and realize self-repair function [67]. Moreover, the introduction of superhydrophobic coatings has played a synergistic role in strengthening multifunctional wood and prolonging its service life [21]. In recent decades, people have become more and more familiar with micro-nano structural materials. It is found that micro- and nanomaterials have excellent hydrophobicity [68,69], high thermal conductivity [70], self-healing characteristics [71], biodegradability and compatibility [72,73], and have been applied to fields such as chemical coatings [74,75], biomedical devices [76,77], food safety [78,79], and photoelectric device manufacturing [80,81,82]. The combination of wood superhydrophobic coating and micro-nano particles effectively links the latest technology of micro-nano structure and green materials, with wide space for development.

If wood is to be widely used in building materials and other fields, the introduction of a super-hydrophobic coating is inevitable. The preparation of superhydrophobic coatings usually requires the construction of a suitable rough surface, the introduction of low surface energy substances to the rough surface, and further chemical modification of it. However, most preparation processes for superhydrophobic coatings have limitations. Table 1 lists the advantages and disadvantages of the superhydrophobic preparation process in actual production. After comparing the actual production demand, cost, surface durability, hydrophobic effect, and other comprehensive factors. As can be seen from the following table, the sol-gel method has great advantages compared with other methods, such as a lower raw material cost, simple experimental equipment requirements, simple operation and easy control, and a stable and controllable reaction.

Most of the existing reviews only discuss the structure and morphology of the coating on the wood surface prepared by the sol-gel method but have not discussed its mechanical properties, mechanical strength, or other parameters, especially the performance parameters of the super-hydrophobic coating prepared on the modified wood surface under different acid and alkali conditions. In this review, we first reviewed the necessity of wood being superhydrophobic and the advantages and disadvantages of various superhydrophobic preparation processes. In Section 2, the preparation principle of the sol-gel method is described, the structural characteristics and mechanical properties of superhydrophobic coating under different catalytic conditions are discussed, the effect of preparing superhydrophobic coating by the sol-gel method is briefly evaluated and analyzed, and the advantages and disadvantages of the sol-gel method are summarized. Finally, the application and development prospects of superhydrophobic wood coatings are discussed.

## 2. Preparation of Superhydrophobic Wood by the Sol-Gel Method

The sol-gel method is an industrially used method for the preparation of wet chemical materials [103,104], in which the precursor is hydrolysed in the liquid phase and the gel system is obtained by condensation polymerization of the colloidal particles [105,106], and the nanoparticles in the system are adsorbed on the substrate material by physical deposition and form a thin film combined with the surface of the substrate material [107,108,109]. Most of the sol-gel methods use SiO_2_, TiO_2_, Al_2_O_3_, and SnO_2_ as nanoparticle fillers [108,110,111,112], the most important of which is SiO_2_. The reaction precursors corresponding to the nanoparticle fillers, such as ethyl orthosilicate (TEOS) [113,114,115], tetrabutyl titanate [116,117,118], and other polyester organics, are usually selected as compounds with high chemical activity components when the precursors are thoroughly mixed with the reactants and catalysts and polymerised in the solution. The gel is solidified in the system, and the particles are cross-linked into a lattice structure. Subsequent processes such as drying, sintering, and curing [119] allow the nanoparticles in solution to adsorb onto the surface of the material to produce a superhydrophobic coating. The superhydrophobic properties of the coated film can also be further enhanced or functionalised by chemical modification of the coating [120,121,122]. The process of preparing superhydrophobic coatings on wood based on this principle is shown in Figure 1. In the liquid-phase system, the precursor undergoes chemical reactions such as hydrolysis and condensation to form a sol system. After a period of molecular crosslinking, the sol is aged and dried to form a gel system. At a certain temperature, the gel is dehydropolymerized to form a dense coating, i.e., the film to be prepared. The structure of thin film products is closely related to the selection of reaction precursors and drying methods, so solid materials that meet the performance conditions can be obtained by controlling the selection of precursors or chemical transformation in the sol-gel process [123,124]. For the coating prepared by the sol-gel method, dip coating and spin coating are often used to cover the substrate surface in the chemical process. The dip coating method [125,126] is used to immerse the substrate into a well-proportioned sol system and pull the substrate out of the sol at a certain speed. In the process of pulling, the sol will be adsorbed on the substrate, and the coating will be successfully realized. However, in the process of pulling the substrate, the dip coating method will be affected by solvent volatilization and droplet tension, resulting in uneven film thickness distribution, which will affect the experimental effect. Therefore, spin coating is often used in chemical processes. The spin coating method is used to drop the sol on the substrate fixed on the spin coating instrument [127,128], and the rotation of the spin coating instrument makes the droplets spread on the surface of the substrate under the influence of centrifugal force to form a new film [129]. In order to make the substrate board have strong adhesion to the film coating and not be easy to fall off, epoxy resin [130,131] is often used to modify the wood in chemical production to increase the adhesion between the substrate and the film. After drying, the hydrophobic properties of superhydrophobic films should be characterized by tests.

The industrial demand for films in many fields has increased, and the development of sol-gel methods has gradually become diversified. Since Geffchen discovered the raw material metal alkoxy compounds for preparing oxide films [132], the sol-gel method has been received and more attention. Subsequently, researchers realized that oxide coatings, catalytic materials, and micro- and nanoporous materials, which are difficult to produce by traditional processes, can be prepared by the sol-gel method [133,134]. Since then, the sol-gel method has developed rapidly in many fields and is widely used in the preparation of functional coating materials. High-purity nanocomposite coatings, which are prepared by the sol-gel method with certain functional modifications, can be coated on the surface of modified wood, metal, ceramic powder materials, and other substrates by different coating processes to form a film coating with specific functions. In modern research, micro-nanomaterials are increasingly used in the preparation of coating films and have gradually become an ideal choice for applications such as antibacterial organisms, medical engineering, and semiconductors [135].

The structure of the coating obtained by impregnating the substrate under different conditions of the gel system varies. In this paper, the preparation process and application of superhydrophobic films in catalytic systems with different acid-base properties will be described in detail.

### 2.1. Acid Catalytic Conditions

The preparation of silica film materials for wood in colloidal liquid systems usually uses strong acids such as HCl [136,137,138] or organic acids such as citric acid [139,140,141] and oxalic acid [142,143,144] as acid catalysts for electrophile reactions in the system. In the acidic solution system, as shown in Figure 2, the -OH of the water molecule is condensed and substituted with the -OR (-OC_2_H_5_) in ethyl orthosilicate (TEOS), the nucleophile attacks the carbon nucleus of the -OR, the leaving group leaves and is substituted, and the hydrogen proton in the water will chemically combine with the remaining RO-group of TEOS, resulting in a bias in the electron cloud of the central silicon atom. The other side of the -OR group is therefore away from the central silicon atom. The RO-group in TEOS is gradually hydrolysed, the anion in the solution system continues to nucleophilically replace the silicon nucleus due to its greater electronegativity, the water molecule -OH replaces the -OR of TEOS, the atomic electron cloud density decreases, the degree of acid-catalyzed hydrolysis is increasingly reduced, the speed becomes slower, less hydrolysis products are collected in the experiment, and linear SiO_2_ molecules are obtained by separating water or alcohol from the product [145]. Hydrolytic coating in an acid-catalyzed system can form a linear SiO_2_ film layer on the substrate [146,147]. The molecules rely on adsorption to attach to the substrate. Due to the poor intermolecular cross-linking of linear SiO_2_ and the small size of the product particles obtained from the hydrolytic polycondensation reaction, the film formed by the coating prepared using acid catalysis has low wear resistance and poor mechanical properties. In addition, some metal-organic compounds or metal complexes can obtain superhydrophobic surfaces with suitable roughness after chemical modification as opposed to inorganic particles. For example, Lu et al. used inexpensive aluminates modified by the addition of citric acid in their study of sol-gel methods [148]. It is possible to achieve the same effect with complex gels. However, superhydrophobic surface coatings prepared by sol-gel methods are more suitable for chemical process production due to the low contamination and simplicity of the process using inorganic substances.

Under acidic conditions, natural fibers such as cotton and silk can be modified from hydrophilicity to hydrophobicity by the sol-gel method. The Espanhol–Soares team studied the use of different concentrations of citric acid as a catalyst to catalyse the reaction of a mixture of ethyl orthosilicate and ethanol to obtain a superhydrophobic coating after stirring at 60 °C. Performance characterization tests revealed that using citric acid as a catalyst to catalyse the cross-linked esterification bond between the synthetic coating and the surface of the cotton fabric, the samples were found to still have good hydrophobicity, maintaining a contact angle always greater than 150° [149]. Wood, which contains hemicellulose and lignin, is also a cellulose aerogel material, and its surface can achieve certain water repellency after treatment. Kumar’s team achieved the combination of titanium dioxide (TiO_2_) nanoparticles and low carbon chain fatty acids (butyric acid) firstly. Superhydrophobic coatings with a static contact angle of 168 ± 2° and an inclination angle of 6 ± 1° were successfully prepared on the surface of wood. The mechanical and chemical tests show that the mechanical properties of the coating are well characterized, and it can still maintain a certain degree of superhydrophobicity at high impact speed, which can effectively realize the self-cleaning performance of wood surfaces and has good prospects for household applications [150]. In addition to the treatment of TEOS, most experiments also chose to use fluorinated organics for the sol-gel modification treatment. The doping of fluoride promotes mixing between precursors and reactants in the liquid phase, resulting in enhanced reactivity, higher purity, and excellent hydrophobic properties of the resulting thin film materials. Tang et al. used the hydrolytic condensation of silica nanosols, methyltriethoxysilane (MTES), and fluorinated compounds under acidic conditions catalysed by the addition of hydrogen chloride to prepare superhydrophobic surfaces. The developed superhydrophobic surfaces were measured to have a wettability-water contact angle of 166° at low temperatures and maintained excellent superhydrophobicity and anti-freeze properties at freezing temperatures [151]. The introduction of fluoride not only produces a superhydrophobic surface with good properties, but it can also be used to modify cotton textiles using the sol-gel method. Yang et al. used the sol-gel method to prepare fluorinated titanium dioxide sols under acid-catalyzed conditions to obtain superhydrophobic cotton textiles. The superhydrophobic coatings were prepared using acetic acid-catalyzed titanium dioxide sols, which were then modified with free radical polymerisation of fluorinated esters. The fabrics were characterised before and after the modification, and the contact angle of the modified fabrics was found to be 152.5°. The chemical stability of the coated fabrics was tested by immersion in solutions of different acidities and alkalinities, demonstrating the water repellency of the fabrics [152]. Because fluoride not only harms the environment but also produces toxins, it is harmful to the human body when applied to textiles. Therefore, people have explored the preparation process for stable fluorine-free superhydrophobic surfaces. Foorginezhad et al. prepared titanium dioxide hydrosol in an acidic environment by the sol-gel method using titanium tetraisopropyl alcohol and made it into a super-hydrophobic coating by adding polydimethylsiloxane (PDMS). The experimental results show that the water contact angle of the coating is as high as 170° and the sliding angle is less than 10°, which has good superhydrophobic performance. At the same time, the surface flexibility of the fabric covered with the coating has not been affected, and it still maintains good self-cleaning performance. Preparation of fluorine-free environmental protection material capable of realizing oil-water separation [153]. 

As can be seen from the above, in the case of silanes, the sol-gel method under acidic catalytic conditions constructs silica films with alternating links of linear silica molecules. The incomplete hydrolysis of the reaction precursors of the sol-gel method in an acidic environment results in a low reaction efficiency, leading to a low number of condensation products obtained from the reaction. At a given molecular weight of linear silica, the reduced condensation products affect the way the linear molecules stack up with each other, resulting in insufficient intermolecularly linked voids and low film porosity of the product. The reduction in intermolecular voids in the film tends to refract external light, resulting in a film coating with a high refractive index and a tendency to deflect light.

In application, the silica material formed by the acid-catalyzed sol-gel method is mostly a thin film type coating, which is attached to the substrate material by spraying techniques such as the spin-coating method so that it establishes a good connection with the substrate material. The acidic sol-gel method is often used in industry to prepare the ester bonds between cellulose products, which increases the tightness of the connection between the film coating and the textile, helps the film adhere better to the substrate, increases adhesion, and promotes better mechanical properties of the film. In addition, acid-catalyzed films can be modified by chemical treatment to obtain films with excellent physical properties, such as superhydrophobicity, piezoelectricity, light transmission, etc., making the preparation of coatings more complete and convenient for use in everyday applications.

Acid catalysts are one of the most important raw materials in the preparation of superhydrophobic thin film materials, such as hybrid films. With the continuous development of thin film materials, thin films with superhydrophobic properties have also been found and put into use in emerging technical fields, which has become a hot topic for people to study. For example, superhydrophobic films with high transparency have been applied in solar cells, automobile glass [154], and traffic indication systems [155], which can not only enhance optical performance but also stably cope with temperature changes and rain with a certain speed. In addition to their application in the automobile industry, superhydrophobic film materials are also used in microfluidic controlling systems [156].

The superhydrophobic surface is self-cleaning, with water droplets rolling off the surface and carrying away particles of impurities from the device, which can relieve the requirement that the machines need specific mechanical cleaning and can realize the function of self-protection of the machine. Usually, when preparing thin film materials, the first choice is the economical and environmentally friendly sol-gel method, which should be carried out in a low-temperature liquid phase using catalysts to provide the reaction environment. The reaction process is easy to control, and the products are evenly dispersed in the mixed system. However, in the process of aging and drying the gel, the film will be uneven in stress and crack due to the volatile contraction of the internal liquid. Improving the toughness of the film and adjusting the ratio of the substances in the liquid phase are important issues in the preparation of thin film materials by the sol-gel method.

### 2.2. Alkaline Catalytic Conditions

Catalytic systems with different acidity and alkalinity have different mechanisms of action on the hydrolysis of TEOS, leading to large differences in the structure of the resulting hydrolysis products [157,158]. Under the catalysis of alkaline solution, the silicon atom of TEOS is directly subjected to nucleophilic substitution by the -OH of the water molecule in solution; the electron cloud of the attacked group is shifted and favours the -OR group on the other side when the silica-oxygen bond is weakened to break and substituted by the -OH of the water molecule. The exothermic reaction leads to an accelerated rate of hydrolysis of the group. As the rate of condensation of TEOS is greater than the rate of hydrolysis, the full reaction produces larger particles of dimeric molecules, i.e., silica particles, which are approximately spherical in shape and contain a large number of hydroxyl groups on their surface. Under alkaline catalytic systems, TEOS tends to hydrolyse completely, producing a loose film of spherical particles of silica [159]. However, its adhesion and mechanical strength are relatively poor. 

In Figure 3, the catalytic process of the sol-gel process under alkaline conditions based on the above principles is listed (1–3), starting with the nucleophilic and electrophile reactions in the colloidal system, where hydroxyl groups in water react with the silane by substitution. Then, the presence of alkali in the solution system can play a facilitating role in the hydrolysis of the silane, promoting the reaction in the positive direction and making the silane completely hydrolysed. The hydrolysis products gradually accumulate in the solution, and finally, the products obtained from the hydrolysis of the silane undergo a simple polymerisation reaction of dehydration and dehydration to obtain multiple condensation molecules. The final process is the simple polymerisation of the products obtained from the hydrolysis of silane by dehydration and dehydration to give multiple condensation molecules.

The most common alkaline catalysts are NH_4_OH [160,161,162], NAOH [163,164], and silicates [165,166], which are selected for their superhydrophobic coating effect at a suitable pH of 9–12 to facilitate the chemical reaction [167,168]. Rao et al. prepared highly hydrophobic silica films by the sol-gel method using an alcohol solution containing silica precursors (methyltrimethoxysilane (MTMS), methanol (MeOH), and ammonium hydroxide (NH_4_OH) by dip coating. By using sodium hydroxide as a catalyst for the preparation and providing an alkaline environment for the colloidal system, the system was dried, dip-coated, and then silylated. The hydrophobic silica films prepared under alkaline conditions were tested to achieve a modified surface water contact angle of 166 ± 2° and a roll angle of less than 5°. The films were mechanically characterised and found to have high water repellency and mechanical stability [169]. Wang et al. prepared a superhydrophobic wood coating of SiO_2_ nanoparticles on the surface of wood by the sol-gel process. The ultrasonically cleaned wood samples were immersed in a mixture of TEOS and ethanol with NH_4_OH as the catalyst, and after being placed at room temperature, cleaned, dried, and modified by adding PTOS, the contact angle of the sample wood surface was tested and found to be 164° with a sliding angle of less than 3°, meeting the expected requirements of the experiment [170]. Most modified superhydrophobic woods have poor durability. In order to improve the modified superhydrophobic wood films. Jia et al. prepared highly abrasion-resistant superhydrophobic woods using SiO_2_ nanoparticles and vinyltriethoxysilane (VTES) by a catalytic drive with alkali NAOH. The wood was treated by placing it in a mixture of sodium hydroxide solutions at a concentration of 0.1 mol L^−1^. Samples were obtained by heating the reaction and tested to obtain wood with a water contact angle of 156.6° and a sliding angle of 1.8°. The addition of NAOH catalysed the condensation reaction of VTES in the gum liquid system. This helped to graft a large number of hydrophobic silica nanoparticles onto the surface of the wood, providing good mechanical and wear-resistant properties for the superhydrophobic coating of the wood [171]. Wang et al. prepared superhydrophobic wood surfaces by mixing solutions made from modified silica particles with polymer emulsions in an alkaline environment. The experiments used ammonia as a catalyst to provide an alkaline system to catalyse the reaction between silane reagents and anhydrous ethanol. Superhydrophobic wood surfaces with a water contact angle of 153 ± 1° and a sliding angle of less than 5 ± 0.5° were obtained after full reaction within the solution, ageing and evaporation, and drying. The characterisation shows that the prepared coatings have good superhydrophobicity and mechanical stability in solutions or solvents of different acidity and alkalinity [172].

The alkaline catalytic conditions promote the binding of silica spherical particles with hydroxide groups, which is more conducive to the modification of silica particles and the preparation of superhydrophobic coatings. When silanes are used as reaction precursors, alkaline solutions provide accelerated catalysis for the hydrolysis and polymerisation reactions of the sol-gel method. Compared to the inhibition of protonation catalyzed by the acidic system, the alkaline solution promotes the hydroxide substitution process in solution, facilitating the complete hydrolysis of the silane precursor and allowing more hydrolysis products to undergo condensation reactions in the sol-gel polymerisation step, increasing the number of condensate molecules. Due to the complete hydrolysis of the silane in an alkaline environment, the number of condensate molecules is high and the intermolecular spaces are small, all of which accumulate randomly in the form of silica spherical particles to obtain a silica film. As the spherical particles of silica are connected to each other by intermolecular forces, they are poorly cross-linked to each other, resulting in films with low mechanical properties that are prone to tearing.

Except for silica films, silica powder materials can also be prepared in alkaline environments. Most of the methods for preparing silica gel coatings in alkaline systems are obtained by a modification of the Stober method, which uses ammonia as a catalyst to provide an alkaline environment to promote the hydrolysis reaction and polymerisation of silane in alcohol solutions, resulting in a powdered material of monodisperse silica particles. The powder material and the film are modified with chemical substances to compensate for the poor film adhesion and low mechanical strength caused by the voids between the spherical particles of silica.

Compared with the acidic catalytic conditions, when preparing superhydrophobic coatings on wood surfaces by the sol-gel method, people usually choose alkaline solvents as catalysts. As the coating structure prepared in an alkaline environment is mostly powder material, the particle size belongs to the micro-nano structure, and the particles are uniformly distributed. The surface appearance of the formed superhydrophobic coating is compact, and the preparation process is easy to operate. However, during the operation, it was found that the particle size of the product was different, the particles contained more impurities, and the superhydrophobic surface roughness decreased. It is also a difficult problem to solve because of the large gap and poor mechanical properties of particles caused by alkaline catalysis. Therefore, the selection of a green-friendly catalyst and suitable substrate, the optimization of catalytic preparation process, and the accurate control of the superhydrophobic appearance and mechanical properties of the substrate surface are important development paths for preparing modified wood superhydrophobic surface coatings by the sol-gel method.

### 2.3. Others

In addition to the two catalytic conditions described above, the performance and yield of the coating can be adjusted by changing the reaction conditions and reactants [173]. The addition of TEOS sols prepared in an acidic environment to an alkaline solution system, which is adjusted to have a suitable pH level, increases the content of linear silica molecules obtained by acid catalysis. Due to the large number of hydroxyl groups attached to the surface of the spherical silica particles obtained by alkali catalysis, the linear silica molecules can be uniformly distributed in the spherical silica particles. When the prepared hybrid sol was coated on the surface of the substrate, it was found that the linear silica molecules would randomly fill in between the voids formed by the silica spherical particles, increasing the adsorption of the coated film to the substrate material and improving the wear resistance and mechanical properties of the film layer. Wu et al. dip-coated graphene and TiO_2_ nanoparticles onto a PLA-based material using the sol-gel method. Upon probing, it was found that the coating prepared by adding a mixture of alkali and curing agent for catalysis could convert the PLA film from a hydrophilic film to a superhydrophobic biomaterial, and the water contact angle of the superhydrophobic material coating was found to be 150° upon characterisation [174].

Furthermore, the sol-gel method can also realize the preparation of the film coating with the self-transformation of hydrophilic and hydrophobic functions. Shi et al. prepared TiO_2_ superhydrophobic films on filter paper using the sol-gel method, which was modified by the hydrolysis of octyltrimethoxysilane to obtain samples with a sliding angle of 5° and a contact angle of 150°. After exposing the samples to UV light for 20 h, the coating changed to a hydrophilic film, and after leaving them in the dark for a fortnight, the coating would again change to a hydrophobic film [175]. Tadanaga’s team found that continuous exposure of TiO_2_, a photosensitive material, to changing light causes branched chains on thin layers of TiO_2_ to break and that an increase in the amount of low-free-energy material can alter its hydrophobicity [176]. In addition, silica nanoparticles have anticorrosive properties. By conducting research into MTES, Uc-Fernández’s team synthesised functional nanoparticles, namely MCM-41 silica particles (MCM-41-HDTMS), by using cetyltrimethoxysilane (HDTMS), and doped them with MTES to construct a composite superhydrophobic coating. Experimental characterisation and testing revealed that the functional nanoparticles added to the matrix were deposited on the surface of the samples, and the surface contact angle increased to 155°, transforming from a hydrophobic to a superhydrophobic surface. The functional nanoparticles located on the surface facilitated the transfer of electronic charge to the surface of the coating, providing some protection to the substrate beneath the coating. Furthermore, the silica particles provide the coating with an anti-corrosion degradation function [177].

As can be seen from the above, in the preparation of superhydrophobic coatings by the sol-gel method, changes in the acidity of the solution reaction system, the material of the reactants, and the reaction conditions can affect the effect of the prepared superhydrophobic coated films. The internal structure of the film is thus altered, which in turn affects the connection between the coating and the substrate material and the tolerance of the mechanical properties. When silane is used as the reaction precursor, silica films can be prepared by acidic or alkaline solutions and attached to the surface of the material to form a superhydrophobic coating. When carbon dioxide nanoparticles are used as the reaction substrate and titanate is used as the reaction precursor, titanium dioxide films can be prepared by the sol-gel method with a photographic effect. This can be used in a variety of industrial applications.

The properties of the prepared coatings can also be altered by changing the nanoparticles deposited in the sol-gel. After studying micro-nanoparticles, it was found that titanium powder has optical properties, copper nanoparticles have certain anti-corrosive properties [113], and silver nanoparticles can be effective in removing bacteria [101]. Nanoparticles in the form of reactants are prepared as a superhydrophobic film on the surface of wood by the sol-gel method, and after some chemical modification, a superhydrophobic coating material with composite properties can be achieved.

## 3. Discussion

With the promotion of global peak carbon dioxide emissions and carbon neutrality, it is imperative to use environmentally sustainable and eco-recycling materials in chemical process manufacturing. Wood has attracted great attention because of its advantages, such as its wide sources, low cost, and green recyclability. Wood is modified by chemical means to have more properties that meet the needs of production and life on the premise of good mechanical properties and elasticity. In this paper, the development process of preparation technology and the sol-gel method for superhydrophobic surfaces is summarized, which embodies the preparation principle of catalyst for superhydrophobic surfaces, emphasizes the development problems of the sol-gel method at present, and discusses the research status and challenges of the sol-gel process for superhydrophobic coating on modified wood surfaces. As described in this paper, the sol-gel process is green, simple, and suitable for large-scale industrial production, but it has the disadvantage of low preparation efficiency in the case of acid catalysis and poor physical properties and mechanical strength of the surface coatings prepared in the case of alkaline catalysis. Using acid catalyst and alkaline catalyst together in preparation can effectively realize complementary advantages, but compared with industrial large-scale preparation and application, the yield problem has not been solved perfectly. 

## 4. Outlook

Large-scale process production should be the necessary way to develop the preparation process for superhydrophobic surfaces. From the perspective of chemical production, the sol-gel method is superior to other preparation methods in terms of raw material cost, operation process, stability of performance, and process pollution. The sol-gel method is not selective in terms of substrate requirements, and substrates of all sizes and shapes can be prepared. In comparison, the sol-gel method has the future development potential of mass production. The research on superhydrophobic surfaces began with the imitation of the structure of natural organisms adapting to the environment in the early days, which gradually developed to prepare hydrophobic coatings with different functions on the surfaces of various substrates and has been extended to be combined with micro-nanomaterials to modify and design the required surface structure. Especially through the research of micro-nano materials, the coating surface with hydrophobic and hydrophilic properties can be designed to adapt to environmental changes. The superhydrophobic surface appearance is becoming increasingly dense and highly porous. Accordingly, the application field of superhydrophobic surfaces has also expanded from oxide film preparation to biomedicine, optical devices, oil-water separation, drug delivery, and other directions, which has brought significant changes to the chemical industry, and the national defense industry. Multifunctional wood with a superhydrophobic surface is becoming a key research field for researchers at present. With the development of sol-gel technology, the problems in industrial production are gradually exposed. For example, how to efficiently increase the yield of superhydrophobic coatings, improve the performance parameters of the coatings in the process of large-scale production, avoid environmental pollution caused by using fluoride to modify low surface energy substances, and reduce equipment costs under environmentally friendly conditions are all issues that require our careful consideration and efforts to improve. Herein, we make the following prospects:(1)People should continue to study the preparation of functionalized superhydrophobic surfaces. Self-cleaning is the basic characteristic of a superhydrophobic coating due to its unique hydrophobic angle and structure. Based on this characteristic, the coating or substrate can be modified to make it functional. For example, in the multi-directional development of anti-corrosion, anti-reflection, anti-icing performance, sterilization, and magnetic response [178], it can adapt to the changes in the application environment in different fields better, such as military and national defense. Of course, if the micro-nanostructure covered by the superhydrophobic coating is modified to make it transparent, it will further realize the excellent performance of high light transmission and less energy consumption and further expand the application scope. The piezoelectric substrate under the hydrophobic surface can achieve the effects of hydrophobicity and power generation, and the materials made in this way can be well developed as green energy.(2)In order to enhance the mechanical effect and performance parameters of superhydrophobic surfaces prepared by the sol-gel industry, the study of superhydrophobic surfaces with shape memory functions provides a new development idea [179]. The shape-memory superhydrophobic surface can realize the shape switching and finally return to the original shape under the induction of external light, magnetism, temperature, and other factors. Dynamically controlling the micro-nanostructure on the surface of the substrate can be extended to the study of self-healing superhydrophobic surfaces. The recovery ability of the induced response materials with shape memory functions is higher than that of other materials. When receiving external stimuli, the damaged parts can self-repair to their initial state and completely restore their superhydrophobic properties. The super-hydrophobic surface with shape memory function enhances service life and reliability, which can solve the problem of high brittleness due to poor mechanical properties. However, the self-healing ability of shape memory materials is still at an early stage of research, and it is still a difficult challenge to prolong their service life while maintaining a certain degree of mechanical effect.(3)In the sol-gel process, to meet the needs of different organic solvents in the catalytic environment, the selection of solvents and modifiers should be further studied. Organic solvents are volatile, which not only pollutes the environment but also does some harm to researchers’ health. As for the selection of modifiers that can reduce the surface energy, fluoride is selected for experiments in most studies. However, fluoride is not only harmful to the environment, but the toxicity of the fabric has not been reduced when it is used in the modification of textiles. Modification treatment can be carried out by changing temperature or electrochemical assistance. The significance of the selection of experimental reagents for the environment should not be underestimated. Only environmentally friendly reagents can be better used for large-scale industrial production.(4)For the coating problem between coating and substrate, when nanoparticles (such as silica) selected by spray inlay are used to bridge the substrate wood, the combination and adhesion between them will also be affected, which will reduce the mechanical resistance of the superhydrophobic surface and make the surface easy to peel off. At present, most of the preparations use small-sized base boards in the laboratory, which are not suitable for industrial process production. We should explore ways to maximize the utilization rate of wood and reduce the loss of waste in production.(5)Aiming at the problem of high energy consumption in the sol-gel process, we should try our best to study new green energy sources in order to reduce the cost and energy consumption in industrial production and realize green and environmentally friendly large-scale production.

## Figures and Tables

**Figure 1 ijms-24-09675-f001:**
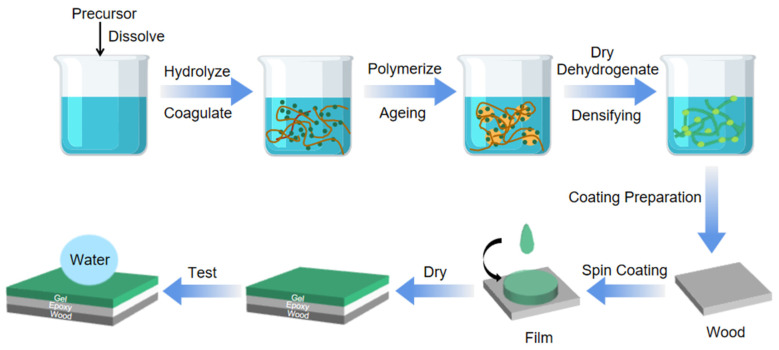
Preparation of superhydrophobic coatings by the sol-gel method.

**Figure 2 ijms-24-09675-f002:**
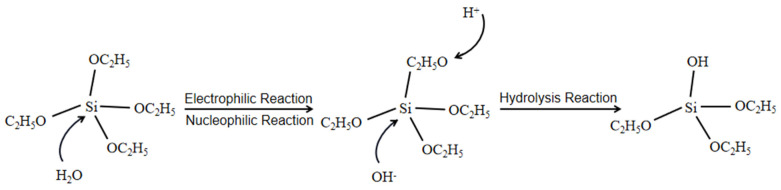
Mechanism of the acid-catalyzed reaction by the sol-gel method.

**Figure 3 ijms-24-09675-f003:**
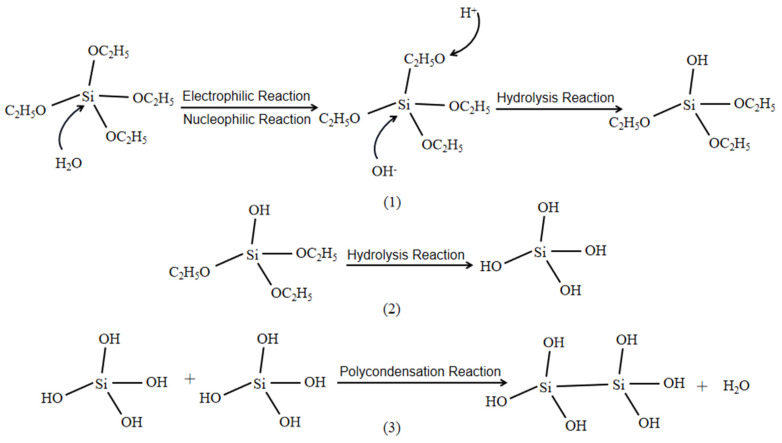
Mechanism of the alkali-catalyzed reaction by the sol-gel method. (**1**) the hydrolysis process of TEOS partial groups, (**2**) the complete hydrolysis process of TEOS, (**3**) the process of dehydration and polymerization between products when TEOS is completely hydrolyzed.

**Table 1 ijms-24-09675-t001:** The advantages and disadvantages of the superhydrophobic preparation process in actual production.

Method	Materials	Principle	Advantages	Disadvantages	Refs.
Wet Etching Method	Strong acids, bases	Chemical reagents soak, React with the etching solution	Stable, Suitable for mass production	Reagent destruction, Environmental pollution	[83,84,85,86]
Dry Etching Method	Laser Plasma	Bombarding the surface of the substrate to cause it to corrode	Simple operation, High etching accuracy, Controlled reaction	Expensive equipment, High energy consumption, Laboratory stage	[87,88,89]
Stencil Printing Method	PDMS, Other templates	Template secondary transfer replication technology	Simple to handle, Easy to obtain, Low cost	Substrate deformation, Making mass production, Difficult	[90,91]
Graft Copolymerisation	Polymer hydrogels, Polymers	Chain transfer grafting, Radiation grafting	Stable performance, Long-lasting modifications	Long-term preparation, high costs	[92,93,94]
Layer-by-layer Self-assembly Method	Polyelectrolyte	Multi-layer alternating deposition technique	Long lasting, Low cost, Simple to use	High toxicity, Complex processes, Unsuitable for large substrates	[95,96]
Hydrothermal Method	Hydrothermal reagents, Base metals	Dissolution, Recrystallisation processes at high temperature and pressure	Good mechanical stability, Environmental-friendly	Poor safety, High equipment requirements	[97,98,99]
Sol-gel Method	Nanoparticles, such as metal salts and other polymer active compounds	Condensation polymerisation reactions under colloidal liquid systems	Mature, Mild, Easy reaction system	Poor product performance	[100,101,102]

## Data Availability

Not applicable.

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
