# Peer review of "Advances in Sol-Gel-Based Superhydrophobic Coatings for Wood: A Review"

_ijms, 2023, doi:10.3390/ijms24119675_

Round 1

Reviewer 1 Report

·  The term biomimetic is only mentioned in the Introduction and in the Summary and Outlook of the manuscript, which makes it redundant in the title and as the keyword.

·    The abstract should briefly state the main conclusions.

·  This paper simply summarises the current state of knowledge on the subject of biomimetic superhydrophobic coatings for wood. However, the manuscript lacks a critical evaluation of the topic.

·  The authors should also comment on related review papers published on this topic and how their manuscript provides new insights and perspectives that go beyond the existing reviews.

·       Authors should not use an excessive number of citations to support one point and should ensure that their citations are accurate. The quoted statements do not always support the statements made in the manuscript. For example, ‘The anisotropic nature of the vertically aligned channels in its internal microstructure [17-20] gives the wood a certain insulating potential [21-23], which makes it an ideal alternative to glass [24,25].’ Ref.17 self-citation and it does not support the point the authors wish to make.

Reviewer 2 Report

As the title suggests, the literature review should focus on improvements or innovations in the production of superhydrophobic sol-gel coatings for wood protection. But even from the abstract it is not clear what the review is aimed at (consideration of methods for achieving superhydrophobicity? There are many similar reviews.) The question arises: what is the uniqueness of this review? In this edition, I cannot recommend a draft for publication.

In addition, I attach a number of comments that can improve this draft:

1) The authors added the word "review" to the keywords! Why?

2) L 43-45 repeated sentence (the text must be well proofread before being sent to the editor).

3) F1 - there is no description for the picture (incomprehensible stages). Is this an original drawing? Or you should post a link.

4) Working with literature (from what I noticed): some literary sources (119, 121, 122) do not correspond to the text of the manuscript (the text in the draft distorts the data in the articles). For example, the authors of the review write "achieving good superhydrophobic properties" (L145), although the contact angles are much lower in the original article.

Also, the term "good superhydrophobic or superhydrophobicity" should not be used, it shows the personal attitude of the authors (for whom is it good and why?).

Did the authors come up with the term “art etching” themselves?

5) The conclusions resemble the introduction, they are poorly written. For some reason, the authors gave titles to the paragraphs of the conclusions, this should not be done.

6) How does the sol-gel method differ from others? Exactly what is better? It would be interesting to see a comparison of the stability of the superhydrophobic properties of different coating methods.

7) In summary: The review does not show the analysis and evolution of the sol-gel method for wood modification. What is new and what has been achieved?

Reviewer 3 Report

The submitted paper is focused on sol-gel methods to prepare superhydrophobic surfaces, in particular on the preparation process and application of superhydrophobic films in catalytic systems with different acid-base properties, predominantly on the basis of silicon compounds. With this respect, maybe this should be more stressed in the abstract and also the introduction could be adapted with this respect. Even the title could reflect better this specific topic.

Some other comments can be found in the attached file.

Round 2

Reviewer 2 Report

The authors revised the text of the draft, but the main idea of the work remains unclear to me (What did the authors want to discuss?). There are quite a lot of reviews in the scientific literature on the basics of the sol-gel method (https://doi.org/10.1155/2021/5102014; doi: 10.11605/j.pnrs.201802008; DOI: 10.1039/C5MH00260E), its use for obtaining superhydrophobic coatings (https://doi.org/10.3390/ma14226799; https://doi.org/10.1007/s10971-016-4027-y) and even for wood surface protection (https://doi.org/10.1007/s00226 -021-01290-w; 10.1039/D2TA09828H; https://doi.org/10.1007/s00226-003-0205-5; DOI: 10.7569/RAA.2013.097308). The title of this work "Advances in Sol-gel Based Superhydrophobic Coatings for Wood: A review" suggests that the latest innovations and features of the application of the sol-gel method for wood modification will be reviewed and discussed. Unfortunately, this is not at work.

I see no reason to publish a typical review, there are similar works in the scientific literature. This draft needs to be thoroughly revised, in this edition I can not recommend for publication.

Reviewer 3 Report

I would like to thank to the authors for respecting my comments and suggestions in the previous review round. I think this is now a nice review of a specific field of research.

Author Response

We are very grateful to you for your encouragement and valuable advice on our work.